# Hospital Variation in Feeding Jejunostomy Policy for Minimally Invasive Esophagectomy: A Nationwide Cohort Study

**DOI:** 10.3390/nu15010154

**Published:** 2022-12-29

**Authors:** Maurits R. Visser, Jennifer Straatman, Daan M. Voeten, Suzanne S. Gisbertz, Jelle. P. Ruurda, Misha D. P. Luyer, Pieter C. van der Sluis, Donald L. van der Peet, Mark I. van Berge Henegouwen, Richard van Hillegersberg

**Affiliations:** 1Department of Surgery, University Medical Center Utrecht, University of Utrecht, 3584 CX Utrecht, The Netherlands; 2Scientific Bureau, Dutch Institute for Clinical Auditing, 2333 AA Leiden, The Netherlands; 3Department of Surgery, Amsterdam UMC location Vrije Universiteit, 1081 HV Amsterdam, The Netherlands; 4Cancer Treatment and Quality of Life, Cancer Center Amsterdam, 1081 HV Amsterdam, The Netherlands; 5Department of Surgery, Amsterdam UMC location University of Amsterdam, 1105 AZ Amsterdam, The Netherlands; 6Department of Surgery, Catharina Hospital, 5602 ZA Eindhoven, The Netherlands; 7Department of Surgery, Erasmus University Medical Center, 3015 GD Rotterdam, The Netherlands

**Keywords:** esophageal carcinoma, feeding jejunostomy, minimally invasive esophagectomy, complications

## Abstract

The purpose of this study was to investigate hospital variation in the placement, surgical techniques, and safety of feeding jejunostomies (FJ) during minimally invasive esophagectomy (MIE) in the Netherlands. This nationwide cohort study analyzed patients registered in the Dutch Upper Gastrointestinal Cancer Audit (DUCA) that underwent MIE for cancer. Hospital variation in FJ placement rates were investigated using case-mix corrected funnel plots. Short-term outcomes were compared between patients with and without FJ using multilevel multivariable logistic regression analysis. The incidence of FJ-related complications was described and compared between hospitals performing routine and non-routine placement (≥90%–<90% of patients). Between 2018–2020, an FJ was placed in 1481/1811 (81.8%) patients. Rates ranged from 11–100% among hospitals. More patients were discharged within 10 days (median hospital stay) without FJ compared to patients with FJ (64.5% vs. 50.4%; OR: 0.62, 95% CI: 0.42–0.90). FJ-related complications occurred in 45 (3%) patients, of whom 23 (1.6%) experienced severe complications (≥Clavien–Dindo IIIa). The FJ-related complication rate was 13.7% in hospitals not routinely placing FJs vs. 1.7% in hospitals performing routine FJ placement (*p* < 0.001). Significant hospital variation in the use of FJs after MIE exists in the Netherlands. No effect of FJs on complications was observed. FJs can be placed safely, with lower FJ-related complication rates, in centers performing routine placement.

## 1. Introduction

Worldwide, esophageal carcinoma is the 8th most common and 6th most lethal cancer, with its incidence projected to continue to increase [1,2]. Surgical resection, in combination with neoadjuvant chemo(radio)therapy, is the cornerstone of curative treatment. Despite advancements in minimally invasive techniques and early recovery after surgery (ERAS) protocols, postoperative morbidity of esophagectomy is still reported in around 60% of patients, and mortality is reported in around 2–3% of patients [3,4].

Anastomotic leakage of the esophagogastrostomy is one of the most severe complications. Historically, the nil-per-mouth policy has been used to reduce the risk of anastomotic leakage and aspiration pneumonia [5]. ERAS protocols, however, stress the need for good nutritional support by early feeding, with enteral feeding as the superior choice [6,7]. Malnutrition greatly increases the rate of postoperative complications and subsequently slows postoperative recovery [8,9]. Esophageal cancer patients frequently present malnourished due to dysphagia, and following esophagectomy, their oral intake remains reduced. Therefore, early enteral nutrition is considered crucial in this frail group of patients [10,11]. In the pursuit of reducing postoperative morbidity, the placement of both a naso-jejunal tube and a perioperative feeding jejunostomy (FJ) have been suggested to ensure a nil-per-mouth policy in the early postoperative period and to allow for early postoperative enteral feeding. When postoperative complications do occur, a major advantage of naso-jejunal tubes and FJs is the ability to ensure enteral feeding without the need for an intervention.

However, both feeding routes come with disadvantages. A naso-jejunal tube causes great discomfort for the patient, while FJ placement as an additional procedure is associated with complications [6,12]. Minor complications, such as catheter occlusion and dislodgement, are reported in around 13–38% of cases [9,12,13]. These problems with FJs have been reported to be the most frequent reason for return visits to the emergency department after esophagectomy [14]. Major complications, such as small bowel obstruction and jejunal torsion, occur less frequently and are reported in around 0–17% of patients, but may require invasive treatment and even lead to death [9,12,13].

Currently, no consensus exists in the literature regarding the feeding route and the routine placement of FJs after esophagectomy for cancer. Hospital variations regarding protocols indicating the need for FJ and varying placement techniques have prevented the implementation of national consensus guidelines. When placed, different surgical techniques can be applied, which may lead to differences in complication rates. The additional procedure securing nutritional support may also impact patient outcomes. The aim of this study was to investigate hospital variation in placement, surgical techniques, and safety of FJ during minimally invasive esophagectomy (MIE) in the Netherlands.

## 2. Materials and Methods

### 2.1. Study Design

This retrospective, nationwide cohort study used data from the mandatory Dutch Upper Gastrointestinal Cancer Audit (DUCA). In the Netherlands, all patients undergoing surgery with intent of resection for esophageal and gastric cancer are registered in this database. All postoperative outcomes are registered until discharge or 30 days after surgery. Under Dutch law, no informed consent or ethical review is required, as patient and hospital data are registered anonymously. This study was approved by the DUCA scientific committee (DUCA202104).

An online questionnaire was conducted among all Dutch hospitals performing esophageal cancer surgery in in 2021. The questionnaire included 11 questions regarding routine placement of FJ and preferred surgical (entry) techniques, anti-rotation stitches, fixation, and feeding protocols, which are shown in Appendix A. From each unit, one lead surgeon completed the questionnaire on behalf of each hospital.

### 2.2. Patients

All patients registered in the DUCA with esophageal cancer who underwent curative MIE, followed by a gastric conduit reconstruction, between January 2018 and December 2020 in a center that is currently (as of 1 January 2022) performing MIE were considered for inclusion. This relatively small time frame was selected to enable accurate comparison of the questionnaire results with the DUCA data. Patients undergoing salvage esophagectomy and those with missing FJ data were excluded.

### 2.3. Outcome Measurea

The primary outcome measure was the percentage of patients per hospital in whom an FJ was placed during, before, or shortly after MIE, and who were fed using the FJ. Secondary outcome measures were the FJ policies and techniques for each hospital, as measured by the questionnaire. Further secondary outcome measures were postoperative outcomes, shown in Appendix A, compared between patients with and without FJ. Other secondary outcome measures were the percentages and Clavien–Dindo scores regarding FJ-related complications and the resulting reinterventions.

### 2.4. Variables for Analyses

The secondary outcome measures described above were compared between centers that routinely placed FJs and those that did not routinely place FJs. When a hospital had an FJ placement rate of over 90%, it was considered to routinely place an FJ. The 90% cut-off was established in an expert consensus meeting to which all Dutch upper gastrointestinal surgeons were invited.

### 2.5. Statistical Analysis

Each patient in the dataset was assessed for FJ placement. The hospital variation in FJ placement rates was investigated using case-mix corrected funnel plots [15,16]. Multivariable logistic regression analyses estimated (E) FJ placement rates per hospital based on the patient, tumor, and treatment characteristics: sex, age, weight loss, body mass index (BMI), Charlson Comorbidity Index (CCI), American Society of Anesthesiologists (ASA) score, previous esophageal or gastric surgery, tumor location, clinical tumor stage, clinical node stage, and histology (categories are shown in Appendix A). A case-mix corrected funnel plot showed the E on the x-axis and the actual FJ placement rate (observed = O), divided by the E (observed/expected ratio) on the y-axis. An O/E ratio larger than 1.0 indicates more observed FJ placement than expected, based on a hospital’s case-mix, whereas an O/E ratio smaller than 1.0 indicates fewer than expected FJ placements. The 95% confidence intervals were computed around the benchmark (O = E).

The X^2^, or Fisher’s exact test, was used to compare the patient, tumor, and treatment characteristics described in Appendix A between patients with and without FJ in all hospitals and those in hospitals not routinely placing FJs. Univariable and multilevel multivariable logistic regression were used to compare the secondary outcomes between patients with and without FJ. All factors in Appendix A were used in the multivariable models. The hospital identification number was added to the model as a random effect to correct for unmeasured hospital differences, in case the log likelihood-ratio test showed a better fit compared to the original multivariable model. In case of an insufficient degrees of freedom for the entire correction model (i.e., <10 events per category in the multivariable model), factors with a *p*-value < 0.1 in the univariable analyses were added to the multivariable model. ANOVA analyses were used to estimate the overall *p*-values for the variables. The X^2^ test was used to compare FJ-related complications in centers routinely applying FJ and centers not routinely applying FJ. Two-sided *p*-values < 0.05 were considered statistically significant.

All statistical analyses were performed using R-studio version 1.4.1106, The R Foundation for Statistical Computing [17].

## 3. Results

Between January 2018 and December 2020, 2014 patients underwent MIE for esophageal cancer in 17 hospitals in the Netherlands. Applying inclusion and exclusion criteria resulted in the inclusion of 1811 patients in this study (77.4%) (Figure 1). Two hospitals merged to form 1 new hospital, resulting in 15 hospitals currently performing MIE.

FJs were placed in 1481 (81.8%) patients, with rates ranging from 11.0% to 100% between hospitals (Figure 2). After correction for case-mix variables, the funnel plot (Figure 3) shows 3 hospitals with statistically significant lower than expected placement rates and 2 hospitals with higher than expected FJ placement rates.

### 3.1. Baseline Characteristics

The baseline patient, tumor, and treatment characteristics of patients with and without FJ are shown in Table 1. Patients without FJ had a significantly higher Charlson Comorbidity Index (CCI) and higher unknown T and N stages, underwent a transthoracic esophagectomy more often, had more intrathoracic anastomoses, and were more often treated in a hospital performing more than 40 esophagectomies per year. Patients with and without FJ did not differ significantly in regards to BMI and preoperative weight loss.

The baseline characteristics of patients without and with FJ in hospitals not routinely placing FJs are shown in Appendix A. These groups differed significantly in regards to BMI, ASA score, histology, neo-adjuvant therapy, surgical procedure, anastomotic location, and hospital volume.

### 3.2. Postoperative Outcomes

Multivariable analyses showed that the percentage of patients who were discharged within 10 days (the median hospital stay in the Netherlands is 10 days) postoperatively was higher in the group of patients without FJ compared to patients with FJ placement (64.5% vs. 50.4%, respectively; OR: 0.62, 95% CI: 0.42–0.90). Chyle leakage was more common in FJ patients (9.5% vs. 4.6%; OR: 2.16, 95% CI: 1.27–3.98). However, due to insufficient degrees of freedom, univariable analysis was used for the latter outcome measure. No significant differences were observed in overall postoperative complications and the remaining outcome measures (Table 2).

### 3.3. FJ-Related Complications

Out of 1481 patients with an FJ, FJ-related complications were reported in 45 (3.0%) and severe FJ-related complications in 23 (1.6%) patients. A total of 12 patients (0.8%) required a reintervention under general anesthesia for FJ-related complications. No complications higher than grade IIIb were reported. FJ-related complication rates among hospitals were reported as ranging between 0% and 15.7%.

### 3.4. Routine vs. Non-Routine FJ Placement

Of the 17 hospitals performing MIE, 3 (18%) used an FJ in <90% of patients; these were classified in the non-routine FJ placement group. One hospital where an FJ was placed routinely, but where it was used only selectively, was categorized in the routine FJ placement group. In total, 168/1481 (11.3%) patients were fed through an FJ in the 3 centers not routinely applying FJs. There was a significantly higher FJ-related complication rate in these 3 centers not routinely applying a FJ than in centers that did routinely apply an FJ (13.7 vs. 1.7%, *p* < 0.001). This was also observed for severe FJ-related complications (6.0 vs. 1.0%, *p* < 0.001).

### 3.5. Questionnaire

Appendix A shows the results of the questionnaire among 13 of the 15 Dutch hospitals currently performing MIE. A total of 12 hospitals stated that they perform routine placement of an FJ during esophagectomy, of which 1 hospital stated that the FJ was only used if oral nutritional goals were not met. Only 1 (8%) hospital disclosed that it did not routinely perform FJs after esophagectomy. No clear preferences on abdominal entrance site and type of suture were seen. Purse string sutures were most frequently favored (62%) for fixation of the jejunum to the abdominal wall. A total of 8 hospitals (62%) used anti-rotation sutures, with the majority using interrupted sutures (88%) on at least the distal side (88%). Most hospitals (54%) started feeding over the FJ on the morning after surgery, whereas direct postoperative feeding was initiated in 38% of hospitals. Generally, a step-up feeding protocol was used until full enteral feeding was achieved 2–3 days after surgery.

## 4. Discussion

This study investigated the use and safety of FJs after MIE in the Netherlands. From 2018 to 2020, there was significant hospital variation in the placement of FJs among Dutch hospitals. Most hospitals (80%) routinely place FJs, with placement rates ranging from 11% to 100%. The variation found in the current study regarding hospital FJ placement rates corresponds to the results reported in the current available literature, as no consensus exists on the optimal feeding route after esophagectomy [9].

Early feeding is an integral part of ERAS programs after esophagectomy; however, the optimal route, consisting of direct oral feeding (DOF), total parenteral nutrition (TPN), or feeding with a naso-jejunal tube or an FJ, remains a matter of debate [5,6,18]. Although naso-jejunal tubes, like FJs and TPN, ensure a nil-per-mouth policy, they cause great discomfort to patients [6,12]. Proponents of DOF believe DOF leads to fewer complications and faster time to functional recovery, while opponents express the fear of a slower recovery and higher anastomotic leakage rates [5,19,20]. In the Netherlands, hospitals do appear to prefer the routine placement of FJs.

In this study, a limited effect of FJ on outcomes was observed. Patients with an FJ were discharged later, after the median of ten days, compared to patients without FJ. Similarly, a recent study reported a median in-hospital stay of 30 days for patients with FJ vs. 18 days for patients without FJ [18]. Time to discharge may have been affected by variation in discharge pathways among Dutch hospitals. Additionally, patients are usually discharged with the FJ after esophagectomy in the Netherlands. Patients and their families require training to safely use the FJ, and home care staff experienced with FJs are needed, possibly resulting in extended waiting times before discharge. Some hospitals only apply an FJ in high risk patients, and some FJs were placed secondarily as a result of a complication, potentially influencing results.

Multiple studies reported no or limited differences in postoperative complications and mortality between patients with and without FJs [9,21,22,23]. A recent meta-analysis (4594 patients) investigated the application of FJs [9]. They reported a slightly higher, but still significant, anastomotic leakage and pulmonary complication rate in the FJ group. These results differ from those in our study and might be explained by selection bias, with frailer patients receiving an FJ and insufficient correction for confounders in the individual studies.

Historically, DOF was believed to possibly increase the risk of anastomotic leakage and inadequate caloric intake [24]. Proponents advocate that it stimulates the autonomic nervous system, resulting in fewer inflammatory-based complications compared to other forms of feeding, including FJs [25]. Recent studies have shown no increased prevalence of anastomotic leakage in DOF compared to delayed oral feeding or FJ, along with a faster time to recovery for DOF [20,25,26]. Similarly, our study shows no increased anastomotic leakage rate in the non-FJ group, but it lacks information on time to recovery or nutritional intake, as neither are registered in the DUCA.

An increased postoperative chyle leakage rate was found in FJ patients compared to non-FJ patients (9.5 vs. 4.6%), which was not reported in the meta-analysis (3.2 vs. 3.8%). A recent Dutch study also reported higher chyle leakage rates in patients with FJ compared to those without (10.4 vs. 1.5%) [26]. The higher chyle leakage rate could be caused by the continuous challenge of the bowel with intralipids by tube feeding compared to other forms of feeding. This causes more fat to be digested and subsequently transported through the lymph system, resulting in increased flow in the thoracic duct [27].

An FJ is primarily placed to ensure that enteral nutrition is possible, especially during postoperative complications. Patients on only DOF after MIE have been reported to require the start of nonoral nutrition in 38% of cases, mainly due to complications (36%) [25]. Another recent study identified female sex and higher ASA scores as independent risk factors for requiring an FJ to ensure nutrition [19]. In 29.7% of patients, an FJ was needed due to complications or the failure to meet nutritional goals. The FJ complication rate was reported to be above the rate noted in this study (10 vs. 3%). These findings suggest that there is room for selective FJ placement, as nearly 70% of patients recovered on DOF only. However, a significant group does require an intervention to ensure adequate caloric intake. Further studies should be conducted for identifying the patient group needing nutritional support through FJ and the safety of its routine placement.

This study shows that the use of FJ in the Netherlands is relatively safe. FJ-related complications were reported in 3.0%, and severe complications were reported in 1.6% of patients, with no complications higher than Clavien–Dindo grade IIIb. These numbers are lower than those reported in a recent meta-analysis (10–40% and 2–15%, respectively) [9].

The FJ-related complication rate was higher in hospitals not performing routine FJ placement, for both overall and severe complications. These hospitals did not perform worse in regards to other outcome measures. These results indicate that FJ placement is especially safe in centers performing routine placement. In complex gastrointestinal surgeries, such as esophagectomies, performing more procedures results in better outcome, a phenomenon called the volume–outcome relationship [28]. The differences in this study may be explained by the experience of the surgical team, nurses, and ward doctors show are more familiar with the surgical procedure and FJ care [29,30]. The negative selection of patients receiving an FJ in these centers may also play a role. Importantly, the differences stress the need for national protocols and shared learning in the placement of FJs.

Feeding tube complications are common, and they are reported to be the biggest reason for return visits to the emergency department after esophagectomy [14]. Following surgery, patients in the Netherlands are usually readmitted to the surgical center for postoperative complications. In this study, there was no significant difference in readmission rates for patients with and without FJ. Due to the structure of the DUCA, return visits to the emergency department for feeding tube problems are not registered. Therefore, some of the most common complications of FJs might not be included in this study. Although these complications are rarely severe, nor do they often impact the safety of FJs, they can be a burden to patients.

The techniques used for the placement of FJs varied among hospitals. A form of anti-rotation suturing, which we believe is an important technique for FJs, was applied in only 62% of hospitals. Literature on other placement routines is abundant, although there is no consensus on a gold standard [13,22].

This study has some limitations. Confounding by indication might exist in the current study. Using multilevel multivariable logistic regression analyses, we corrected for known and unknown confounders. However, due to the nature of retrospective cohort studies, the groups were unequal. Hospitals that do not routinely apply an FJ, only place one on indication, potentially causing selection bias.

The DUCA does not register the timing of FJ placement. Therefore, it is unknown whether the FJ is placed before neo-adjuvant treatment, during, or shortly after surgery. Non-FJ-patients with a complication (e.g., anastomotic leakage), in need of extra nutritional support, could have received an FJ after the complication had already occurred. These patients were then registered in the FJ group, while initially not scheduled for receiving one. However, most hospitals perform routine placement during esophagectomy, and no difference was observed in reintervention rates, indicating that this group was small.

## 5. Conclusions

This nation-wide, population-based study showed significant hospital variation in the use and placement techniques of FJs after MIE in the Netherlands. Routine placement is performed in the majority of centers. An FJ ensures adequate enteral nutrition, with no effect of FJ placement on the occurrence of postoperative complications. The additional risk seems limited, as the observed FJ complication rate was only 3.0%. Moreover, the FJ-related complication rate was lower in centers in which FJ placement is a routine practice, showing that the experience that comes with routine placement lowers the complication rate. These results stress the need for a standardized national protocol regarding FJ placement after MIE.

## Figures and Tables

**Figure 1 nutrients-15-00154-f001:**
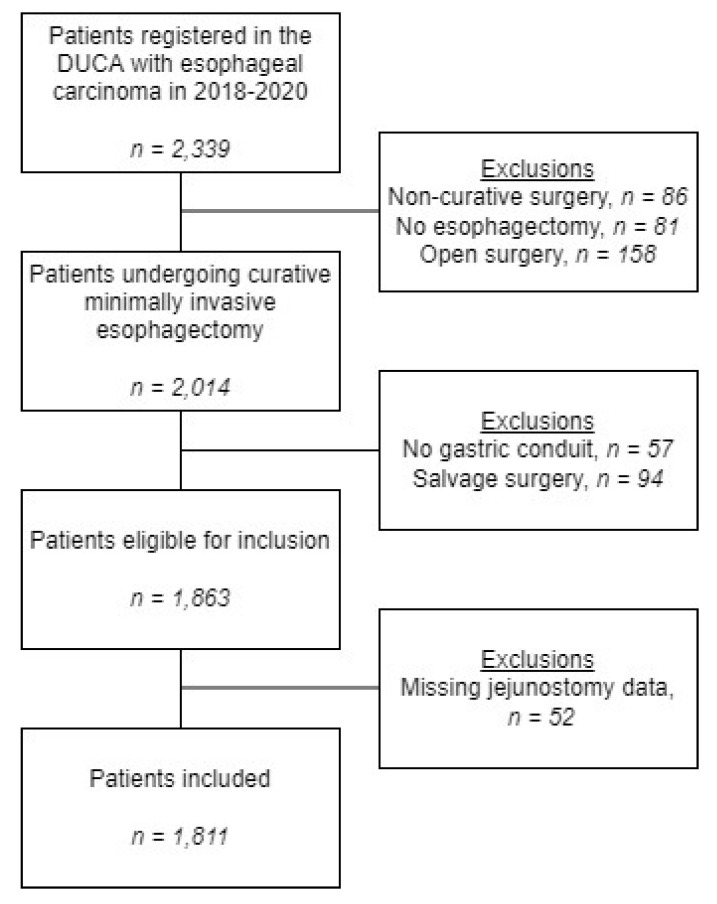
Flowchart of included patients.

**Figure 2 nutrients-15-00154-f002:**
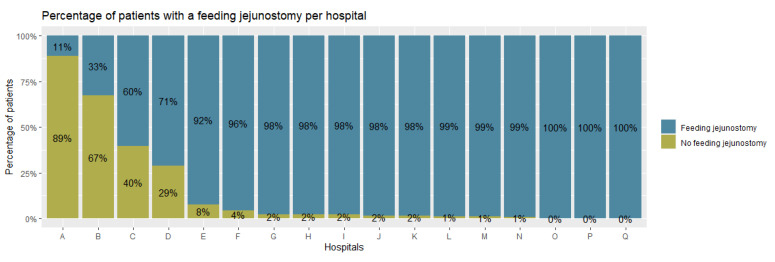
Bar chart showing hospital variation in the placement of feeding jejunostomies, by percentage of patients per hospital, during minimally invasive esophagectomy in 2018–2020.

**Figure 3 nutrients-15-00154-f003:**
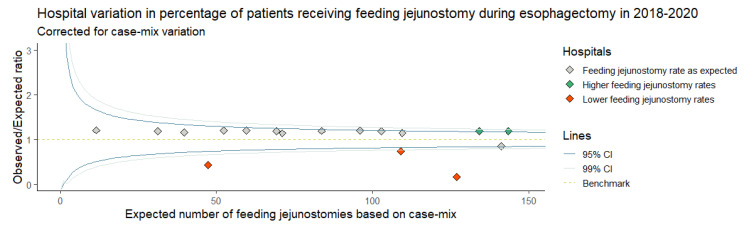
Case-mix corrected funnel plot showing hospital variation in the placement of feeding jejunostomies during minimally invasive esophagectomy in 2018–2020. CI = confidence interval.

**Table 1 nutrients-15-00154-t001:** Patient, tumor, and treatment characteristics of all patients without and with feeding jejunostomies (FJ).

	Patients without FJ *N (%)*	Patient with FJ *N (%)*	Total *N (%)*	*p*-Value (χ^2^/Fisher)
Total	330 (100%)	1481 (100%)	1811 (100%)	
Sex				0.054
Male	243 (73.6%)	1163 (78.5%)	1406 (77.6%)	
Female	87 (26.4%)	318 (21.5%)	405 (22.4%)	
Age in years				0.132
<65	135 (40.9%)	539 (36.4%)	674 (37.2%)	
65–75	163 (49.4%)	748 (50.5%)	911 (50.3%)	
>75	32 (9.7%)	194 (13.1%)	226 (12.5%)	
Preoperative weight loss (kg)				0.150
No weight loss	104 (31.5%)	460 (31.1%)	564 (31.1%)	
1–5	81 (24.5%)	462 (31.2%)	543 (30.0%)	
6–10	84 (25.5%)	324 (21.9%)	408 (22.5%)	
>10	48 (14.5%)	181 (12.2%)	229 (12.6%)	
Missing	13 (3.9%)	54 (3.6%)	67 (3.7%)	
Body Mass Index (BMI) (kg/m^2^)				0.059
<20	14 (4.2%)	102 (6.9%)	116 (6.4%)	
20–25	155 (47.0%)	690 (46.6%)	845 (46.7%)	
26–20	108 (32.7%)	519 (35.0%)	627 (34.6%)	
>30	51 (15.5%)	166 (11.2%)	217 (12.0%)	
Missing	2 (0.6%)	4 (0.3%)	6 (0.3%)	
Charlson Comorbidity Index				0.010
0	126 (38.2%)	673 (45.4%)	799 (44.1%)	
1	72 (21.8%)	371 (25.1%)	443 (24.5%)	
2+	122 (37.0%)	437 (29.5%)	559 (30.9%)	
Missing	10 (3.0%)	0 (0%)	10 (0.6%)	
ASA				0.894
1–2	227 (68.8%)	1007 (68.0%)	1234 (68.1%)	
3+	103 (31.2%)	465 (31.4%)	568 (31.4%)	
Missing	0 (0%)	9 (0.6%)	9 (0.5%)	
Diabetes				0.393
No	269 (81.5%)	1220 (82.4%)	1489 (82.2%)	
Yes	48 (14.5%)	252 (17.0%)	300 (16.6%)	
Missing	13 (3.9%)	9 (0.6%)	22 (1.2%)	
Previous esophagogastric surgery				0.133
Yes	6 (1.8%)	13 (0.9%)	19 (1.0%)	
No	324 (98.2%)	1458 (98.4%)	1782 (98.4%)	
Unknown/missing	0 (0%)	10 (0.7%)	10 (0.6%)	
Tumor location				0.708
Intrathoracic esophagus	256 (77.6%)	1170 (79.0%)	1426 (78.7%)	
Gastro-esophageal junction	72 (21.8%)	298 (20.1%)	370 (20.4%)	
Unknown/missing	2 (0.6%)	13 (0.9%)	15 (0.8%)	
Histology				0.091
Adenocarcinoma	278 (84.2%)	1191 (80.4%)	1469 (81.1%)	
Squamous cell carcinoma	40 (12.1%)	252 (17.0%)	292 (16.1%)	
Other/unknown	8 (2.4%)	29 (2.0%)	37 (2.0%)	
Missing	4 (1.2%)	9 (0.6%)	13 (0.7%)	
Clinical tumor stage				0.015
T0–2	59 (17.9%)	308 (20.8%)	367 (20.3%)	
T3–4	250 (75.8%)	1119 (75.6%)	1369 (75.6%)	
Tx	21 (6.4%)	47 (3.2%)	68 (3.8%)	
Missing	0 (0%)	7 (0.5%)	7 (0.4%)	
Clinical node stage				0.017
N0	125 (37.9%)	552 (37.3%)	677 (37.4%)	
N+	186 (56.4%)	882 (59.6%)	1068 (59.0%)	
Nx	19 (5.8%)	40 (2.7%)	59 (3.3%)	
Missing	0 (0%)	7 (0.5%)	7 (0.4%)	
Neoadjuvant therapy				0.930
None	17 (5.2%)	79 (5.3%)	96 (5.3%)	
Chemoradiotherapy	281 (85.2%)	1267 (85.6%)	1548 (85.5%)	
Chemotherapy	32 (9.7%)	134 (9.0%)	32 (9.7%)	
Other/missing	0 (0%)	1 (0.1%)	0 (0%)	
Surgical procedure				<0.001
Transhiatal	16 (4.8%)	166 (11.2%)	182 (10.0%)	
Transthoracic	314 (95.2%)	1315 (88.8%)	1629 (90.0%)	
Anastomotic location				<0.001
Intrathoracic	246 (74.5%)	856 (57.8%)	1102 (60.9%)	
Cervical	84 (25.5%)	621 (41.9%)	705 (38.9%)	
None/other/missing	0 (0%)	4 (0.2%)	4 (0.2%)	
Hospital volume (esophageal resections per year)				<0.001
≤40	51 (15.5%)	418 (28.2%)	469 (25.9%)	
>40	279 (84.5%)	1056 (71.3%)	1335 (73.7%)	
Missing	0 (0%)	7 (0.5%)	7 (0.4%)	

**Table 2 nutrients-15-00154-t002:** Multilevel multivariable logistic regression analyses of short-term surgical outcomes after minimally invasive esophagectomy compared between patients without and with feeding jejunostomies (FJ).

	FJ Placement	Outcome Incidence (%)	Corrected for ^a^	aOR ^a^	95% CI ^b^	*p*-Value
Overall intraoperative complications (yes)	No Yes	13 (4.3%) 49 (3.5%)	No relevant confounders identified	1–0.81	0.45–1.58	0.510
Overall postoperative complications (yes)	No Yes	180 (59.2%) 901 (64.3%)	All	1–1.27	0.88–1.83	0.199
Severe complications ^c^ (yes)	No Yes	96 (31.6%) 427 (30.5%)	All	1–1.04	0.73–1.48	0.822
30-day/in-hospital mortality (yes)	No Yes	5 (1.6%) 42 (3.0%)	No relevant confounders identified	1–1.85	0.80–5.39	0.198
Chyle leakage (yes)	No Yes	14 (4.6%) 133 (9.5%)	No relevant confounders identified	1–2.16	1.27–3.98	0.007
Anastomotic leakage (yes)	No Yes	40 (13.2%) 259 (18.5%)	All	1–1.36	0.84–2.22	0.210
Pulmonary complication (yes)	No Yes	93 (30.6%) 441 (31.5%)	All	1–1.06	0.82–1.41	0.670
Pneumonia (yes)	No Yes	49 (16.1%) 311 (22.2%)	All	1–1.38	0.90–2.11	0.147
Wound infection (yes)	No Yes	9 (3.0%) 51 (3.6%)	Type of esophagectomy, anastomotic location, and hospital volume	1–0.80	0.39–1.80	0.560
Length of hospital stay (≤ 10 days)	No Yes	196 (64.5%) 706 (50.4%)	All	1–0.62	0.42–0.90	0.013
Prolonged hospital stay (>30 days)	No Yes	24 (7.9%) 139 (9.9%)	No relevant confounders identified	1–1.06	0.57–1.96	0.856
30-day readmission (yes)	No Yes	49 (16.1%) 217 (15.5%)	All	1–0.91	0.65–1.31	0.630
Reintervention (yes)	No Yes	86 (28.3%) 348 (24.8%)	All	1–0.88	0.61–1.27	0.500

^a^ Adjusted odds ratio. Corrected for: sex, age, preoperative weight loss, body mass index (BMI), Charlson Comorbidity Index, American Society of Anesthesiologist (ASA) score, diabetes, previous esophageal or gastric surgery, tumor location, histology, clinical tumor stage, clinical node stage, neoadjuvant therapy, type of esophagectomy, anastomotic location, hospital volume, and hospital as a random effect factor. In case of insufficient degrees of freedom for correction for all possible confounders, only confounders leading to a 10% change in OR were included for analyses. Hospital ID was added as random effect to the model in case the log likelihood-ratio test showed a better fit compared to the original multivariable model. ^b^ 95% confidence interval. ^c^ Clavien–Dindo grade IIIa or higher.

## Data Availability

Data of the DUCA can be obtained under strict rules by filing a research application (www.dica.nl/duca/onderzoek) (accessed on 25 December 2022).

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
