# Peer review of "Hospital Variation in Feeding Jejunostomy Policy for Minimally Invasive Esophagectomy: A Nationwide Cohort Study"

_nutrients, 2022, doi:10.3390/nu15010154_

Round 1
Reviewer 1 Report
This manuscript is well-written and used appropriate statistical analysis to support their conclusion.
Author Response
Thank you for reviewing our manuscript.
Reviewer 2 Report
On request of Nutrients, I have revised the manuscript titled: “Hospital Variation in Feeding Jejunostomy Policy for Minimally Invasive Esophagectomy: A Nationwide Cohort Study”, by Maurits R. Visser MD and colleagues.
With this study, the authors investigated the hospital variation in placement, surgical techniques and safety of feeding jejunostomies (FJ) during minimally invasive esophagectomy (MIE) in the Netherlands. To this end, patients registered in the Dutch Upper Gastrointestinal Cancer Audit (DUCA) that underwent MIE for cancer, were considered. Casemix-corrected funnel plots and multilevel multivariable logistic regression analysis were used to investigate hospital variation in FJ placement rates and to compare short-term outcomes between patients with and without FJ. Additionally, the incidence of FJ-related complications was described and compared between hospitals performing routine and non-routine placement (≥90%-<90% of patients).
The present paper is interesting, and the English language is good. Except for some trivial grammatical error, the paper is well written, and the reading is easy. The design is rational. Anyway, some parts need to be extended and all the manuscript should be checked and corrected to adapt the format to the template offered by Nutrients.
First of all the authors must improve the introduction making reference to a recent work on the topic.
The Role of the Pharmacist in Selecting the Best Choice of Medication Formulation in Dysphagic Patients. J. Pers. Med. 2022, 12, 1307. https://doi.org/10.3390/jpm12081307.
Some examples of issues that need to be addressed:
1) In the title, headings, and sub-headings the first letter of each word should be capital.
2) Sub-headings in Italics should be numbered.
3) Sub-headings in Italics shouldn't have the indentation.
4) Line 22: “were” is incorrect. Please, change it with “was”
5) Fisher with capital letter.
6) The figures captions should be moved after the Figures. In the captions, “Figure” should be in bold and not in Italics (lines 162 and 169).
7) The correspondent author, in the authors list should have the asterisk (*)
8) The authors’ affiliation should have superscript numbers.
9) All abbreviations must be specified at their first mention. Please, check all manuscript and correct where necessary.
10) In the titles of Tables, after “Table” a dot should be inserted in place of colon.
11) The title of Table 2, not in bold.
12) Lines 173, 188, 202, 209 and 218. Please, correct the format of headings according to the template.
13) Pease, remove all spaces between lines that are not necessary.
14) All references do not respect the format required by MDPI journals. Please, correct. Additionally, the references have a double numbering. Please, correct.
15) The conclusions are very poor and need improvement.
1
On these considerations, I suggest Nutrients to reconsider the present work for publication, after minor revisions.
Reviewer 3 Report
Thank you very much to the Editor of NUTRIENTS for allowing me to review the paper entitled ‘Hospital Variation in Feeding Jejunostomy Policy for Minimally Invasive Esophagectomy: A Nationwide Cohort Study.’
Major comments:
The authors suggested that feeding jejunostomy can be placed safely, with lower feeding jejunostomy-related complication rates in centers where it is routine practice. Results were given clearly with sufficient tables and data analysis, but studies are based on relatively outdated literature (9 out of 30 items were published more than five years ago).
Minor comments:
Instead of: X2
It should be: c2
Instead: The X2 or fisher’s exact test was used to compare the patient, tumor and treatment characteristics described in table S3 between patients with and without FJ in all hospitals and those in hospitals not routinely placing FJs.
Should be: The X2 or Fisher’s exact test was used to compare the patient, tumor and treatment characteristics described in table S3 between patients with and without FJ in all hospitals and those in hospitals not routinely placing FJs.
Table 1:
Instead of: BMI
Should be: BMI (kg/m2)
Under Table 1:
Lack of explanation of abbreviations. For instance, CCI, ASA, BMI.
Poor visibility of entries in Figure 2.
Please avoid abbreviations in table titles - for instance, in Table 2 entitled: ‘Multilevel multivariable logistic regression analyses of short-term surgical outcomes after MIE compared between patients without and with FJ’
